# Functional Nutrients to Ameliorate Neurogenic Muscle Atrophy

**DOI:** 10.3390/metabo12111149

**Published:** 2022-11-21

**Authors:** Viviana Moresi, Alessandra Renzini, Giorgia Cavioli, Marilia Seelaender, Dario Coletti, Giuseppe Gigli, Alessia Cedola

**Affiliations:** 1Institute of Nanotechnology, c/o Sapienza University of Rome, National Research Council (CNR-NANOTEC), 00185 Rome, Italy; 2Unit of Histology and Medical Embryology, Department of Anatomy, Histology, Forensic Medicine and Orthopaedics, Sapienza University of Rome, 00185 Rome, Italy; 3Department of Surgery, Cancer Metabolism Research Group, LIM 26-HC, Faculty of Medicine, University of São Paulo, São Paulo 01246-903, Brazil; 4Sorbonne Université, CNRS, INSERM, Institut de Biologie Paris-Seine (IBPS), Biological Adaptation and Aging (B2A), F-75005 Paris, France; 5Institute of Nanotechnology, c/o Campus Ecotekne, National Research Council (CNR-NANOTEC), 73100 Lecce, Italy

**Keywords:** nutraceuticals, natural compounds, muscle wasting, neurodegenerative diseases, sarcopenia, aging

## Abstract

Neurogenic muscle atrophy is a debilitating condition that occurs from nerve trauma in association with diseases or during aging, leading to reduced interaction between motoneurons and skeletal fibers. Current therapeutic approaches aiming at preserving muscle mass in a scenario of decreased nervous input include physical activity and employment of drugs that slow down the progression of the condition yet provide no concrete resolution. Nutritional support appears as a precious tool, adding to the success of personalized medicine, and could thus play a relevant part in mitigating neurogenic muscle atrophy. We herein summarize the molecular pathways triggered by denervation of the skeletal muscle that could be affected by functional nutrients. In this narrative review, we examine and discuss studies pertaining to the use of functional ingredients to counteract neurogenic muscle atrophy, focusing on their preventive or curative means of action within the skeletal muscle. We reviewed experimental models of denervation in rodents and in amyotrophic lateral sclerosis, as well as that caused by aging, considering the knowledge generated with use of animal experimental models and, also, from human studies.

## 1. Neurogenic Muscle Atrophy

Neurogenic muscle atrophy occurs whenever the somatic nervous system is injured or affected by disease, as well as during the physiological process of aging. Diseases that strictly affect motoneurons include amyotrophic lateral sclerosis (ALS), spinal muscular atrophy, and Guillain-Barre syndrome; but other diseases may damage the peripheral nervous system, including poliomyelitis and multiple sclerosis. In addition, denervation of muscle fibers also occurs as an important pathogenic mechanism in muscle wasting associated with cancer cachexia syndrome [1,2].

Without proper trophic innervation, skeletal muscle undergoes remodeling: myofibers shrink in size and some become angular in appearance, compressed by surrounding myofibers, while others relocate the myonuclei in the center of the myofibers [1,3]. At the ultrastructural level, the disorganization of myofibrils and sarcomeres, sarcotubular system, mitochondria, and nuclear localization directly correlates with the time of denervation [4]. In addition to skeletal muscle, surrounding tissues undergo remodeling, including progressive devascularization and the activation of fibro-adipogenic progenitors (FAPs), which contribute to the accumulation of fibro-fatty tissue, in the absence of any significant increase in macrophages and muscle stem cell numbers [5]. Denervation differentially affects slow and fast fibers: differences in the kinetics of myofiber atrophy have been reported in the rat EDL, where type II (fast) muscle fibers are firstly affected, soon after denervation, while type I (slow) muscle fibers resisted atrophy for several months [6]. The histopathological changes of denervated muscles have been extensively reviewed elsewhere [7,8]. In physiological conditions, such as aging, these processes may be reversible [9,10].

### 1.1. Pathways Triggered by Denervation in Skeletal Muscle

Loss of innervation triggers the activation of specific signaling pathways in skeletal muscle, resulting in pathological responses and adaptations, and triggering neurogenic muscle atrophy [5,11,12,13,14]. Upon denervation, skeletal muscle fibers activate three protein degradation pathways: the ubiquitin-proteasome system (UPS), the autophagic/lysosomal pathway, and the Ca^2+^-activated cysteine proteases (calpains). Also, high oxidative stress and redox imbalance are strongly correlated with the extent of neurogenic muscle atrophy, despite controversy regarding an eventual causative role. Similarly, the activation of the AKT-mTOR signaling cascade following muscle denervation and its involvement in neurogenic muscle atrophy are still debated topics. We will briefly review the main signaling cascades triggered by loss of innervation in the skeletal muscle, highlighting the molecular targets that are likely to be modulated by the consumption of functional foods (Figure 1).

#### 1.1.1. Ubiquitin-Proteasome System

The UPS is activated following the denervation of skeletal muscle [15] and consists of a three-step enzymatic reaction cascade, resulting in the covalent attachment of a chain of ubiquitin molecules to the proteins targeted for degradation via the 26S proteasome (reviewed in [16,17]). Three main groups of enzymes orchestrate protein ubiquitination, tagging, and driving to the proteasome for degradation: the E1 (ubiquitin-activating enzyme), the E2 (ubiquitin-conjugating enzyme), and the E3 (ubiquitin–protein ligase). Importantly, the E3-ubiquitin ligases play a key role in this proteolytic cascade since they confer specificity to the process by selectively binding the ubiquitin chain to a protein target.

Among the muscle-specific E3 ubiquitin ligases, Muscle RING Finger-1 (MuRF1) and Muscle atrophy F-Box (MAFbx; also known as atrogin-1) [15], as well as the recently identified MUSA1 [18] and SMART1 [19], are collectively termed “atrogenes”, since their expression is up-regulated upon different atrophic stimuli, mediating the proteasome-mediate muscle protein breakdown. Indeed, knocking out or down their expression protects skeletal muscle from denervation-induced atrophy [15,18,19,20]. Loss of innervation induces E3-ubiquitin ligase expression by the modulation of different kinases, including the p38 mitogen-activated protein kinases (MAPKs) and the protein kinase B (AKT), both converging to regulate the activity of the Forkhead box O (FoxO) transcription factors [21,22,23,24]. In addition, inflammatory cytokines, such as tumor necrosis factor α (TNFα), TNF-like weak inducer of apoptosis (TWEAK), and interleukin 6 (IL-6), induce the activation of NF-kB, Stat3 and FoxO, thus contributing to the activation of the transcriptions of atrogenes [25,26]. Of note, denervation triggers FAPs to persistently activate Stat3 and secrete IL6, thereby promoting muscle trophy and fibrosis [5]. Consistently, inactivation of the Stat3- IL6 signaling in FAPs prevented skeletal muscle from neurogenic muscle atrophy upon acute or chronic denervation, such as in a mouse model of ALS [5]. By altering calcium influx, denervation also activates the Class II HDACs-myogenin axis, leading to the transcriptional activation of both MuRF1 and atrogin-1 genes [27].

#### 1.1.2. Autophagy

Macroautophagy, hereafter referred to as autophagy, is a highly conserved proteolytic pathway which involves more than 40 components coordinating the clearance of damaged proteins and organelles by engulfing cytosolic material into double-membraned vesicles (i.e., autophagosomes) and assisting them to fuse with lysosomes for degradation [28,29]. The family of AuTophaGy-related (Atg) proteins controls the major steps of autophagy: from vesicle nucleation to vesicle elongation (phagophore formation) and closure (autophagosome formation), to the autophagosome-lysosome fusion to form the autolysosome, and the breakdown of its contents [30,31]. Autophagy is constitutively active in all cells to generate energy, to remove damaged organelles, and to support stress resistance. Instead, autophagy imbalance is detrimental for skeletal muscle homeostasis: its hyperactivation contributes to muscle wasting in many diseases [32], while the impairment of autophagy induces muscle degeneration and myopathy-related diseases [33,34,35,36].

The unc-51-like kinase 1 (Ulk1)/Atg1 complex is responsible for the initiation of autophagosome formation by recruiting Atg proteins to the autophagosome formation site, in addition to playing a key role in autophagosome assembly and autophagy flux maintenance [37,38]. The activity of this upstream complex is directly regulated by AMP-activated protein kinase (AMPK) and the mammalian target of rapamycin (mTOR). Indeed, activated AMPK triggers the activation of Ulk1, while the mTOR complex 1 (mTORC1) inhibits it in response to nutrient availability [39,40].

Loss of innervation induces Ulk1 upregulation [41], in addition to enhancing the autophagy flux by increasing the expression of other Atg genes, such as LC3, Atg7, p62, and Beclin1, among others, in an AKT- and FoxO3-dependent manner [42]. According to other authors, instead, denervation reduces autophagic flux in skeletal muscle because of the enhanced activity of mTORC1 [43]. Interestingly, some players, such as FoxO3 and the E3-ubiquitin ligase Trim32, activate both the UPS and the autophagic response upon denervation [42,44,45], coordinately triggering these atrophic responses.

Regardless the pathway involved, autophagy *per se* does not induce neurogenic muscle atrophy in most published studies [43,46,47], even if it apparently plays a role when muscle atrophy is induced in response to denervation concomitant to the activation of the TGFb/HMGB1 pathway [48]. Further studies are required to resolve these contradictory conclusions, which may arise from different experimental settings, such as the study of different muscles and/or different time points analyzed following denervation. 

#### 1.1.3. Calpains

Calpains are activated upon atrophic conditions, including muscle denervation [49], playing a role in the early steps of the sarcomeric protein depolymerization, and the subsequent degradation [50], and in the neuromuscular junction stabilization [51,52]. A direct role for calpain-1 has been demonstrated for desmin filament disassembling in neurogenic muscle atrophy, since calpain-1 down-regulation prevents desmin loss, myofibril destruction and muscle atrophy [50]. Intriguingly, Trim32-mediated ubiquitination of desmin filaments is a prerequisite for its calpain-1-mediated disassembling [50], further highlighting the synergy of the three proteolytic pathways triggered by denervation in skeletal muscle. In addition to calpains, their specific endogenous inhibitors, calpastatins, are modulated in skeletal muscle upon denervation and during aging [53,54,55]. While atrophying muscles display a high calpain/calpastatin ratio, leading to increased calpain proteolytic activity and loss of muscle mass, calpastatin overexpression inhibits the autolytic protease activities of calpains in aging muscles, thereby preventing myofibroillar degradation and sarcopenia, and ultimately increasing the rodent life span [56,57].

#### 1.1.4. Oxidative Stress

Reactive oxygen species (ROS) play an important role in the physiopathology of skeletal muscle by triggering autophagy [58] and apoptosis [59] and affecting myogenic differentiation [60]. Loss of innervation triggers ROS in the skeletal muscle fiber [61,62], concomitantly with a decrease in mitochondrial respiration rate [63]. Despite the observed increase of ROS levels following denervation, oxidative stress (OS) per se is not sufficient to induce neurogenic muscle atrophy. Indeed, treatment of denervated mice with the antioxidant drug Trolox reduced ROS levels but did not prevent skeletal muscle atrophy [62].

Increased OS is one of the known causes of the neurodegenerative disease ALS. About 20% of familial ALS and 5% of the apparently sporadic form are linked to mutations of the superoxide dismutase type-1 (SOD1) gene [64], which encodes for an antioxidant enzyme, pointing to an important role of the OS in the pathogenesis of ALS. Increased amounts of ROS have been observed in the skeletal muscle of mice expressing the human mutant form of the SOD1 gene (SOD1G93A) before motor impairment [65]. Muscle-specific accumulation of the mutant SOD1G93A protein alters redox balance, leading to elevated OS and, consequently, activation of the autophagic pathway and NMJ disassembly [66,67]. Unfortunately, to this point, the treatment with antioxidants has been found to be inefficient to counteract the muscle loss found in ALS patients [68], implying that multiple mechanisms are involved in the onset and progression of this multisystemic disease. 

#### 1.1.5. AKT-mTOR

The mTOR is one of the main regulators of muscle mass in response to different stimuli, and is mostly known for its role in hypertrophy [69,70]. The mTOR complex is a serine/threonine protein kinase that belongs to the phosphoinositide 3-kinase (PI3K)-related kinase family. By interacting with different proteins, mTOR forms two distinct complexes: mTORC1 and mTOR complex 2 (mTORC2). The mTORC1 is acutely sensitive to rapamycin and promotes anabolic processes at the expense of the catabolic ones [71]. In addition to contain the catalytic mTOR subunit and the mammalian lethal with SEC-13 protein 8 (mLST8), the mTORC1 complex is specifically composed of the regulatory-associated protein of the mammalian target of rapamycin (Raptor) and proline-rich AKT substrate 40 kDa (PRAS40) [72,73]. Thus, mTORC1 responds to signaling input along the IRS-1/PI3K/AKT signaling axis, thereby promoting protein synthesis largely through the phosphorylation of two key effectors, p70S6 Kinase 1 (S6K1) and eIF4E Binding Protein (4EBP), overall increasing translational capacity and/or translational efficiency [74,75]. mTORC2, which contains Rictor, is believed to be insensitive to rapamycin and responds to growth factors, thereby mediating cell survival, cytoskeleton organization, and metabolism [71]. Indeed, while mTOR and Raptor muscle-specific knockout murine models show several muscle defects, such as impaired contractile properties, myopathy, fast twitch response, and decreased mitochondrial biogenesis [76,77,78], muscle-specific Rictor knockout mice do not manifest any obvious phenotype, and the inhibition of mTORC2 complex components does not affect muscle integrity, but rather acts on muscle metabolism [76,79].

Unexpectedly, the Akt/mTOR pathway is activated upon denervation in skeletal muscle [80,81]. The activation of the anabolic pathways in atrophying muscles has been considered a paradox for many years, and increasing Akt signaling at the early stages of muscle denervation has been suggested as a potential beneficial response that could be clinically exploited to counteract neurogenic muscle atrophy [82]. Thus, denervation-induced muscle atrophy is not solely mediated by catabolic processes, but it also involves the mTORC1-mediated anabolic pathway, with important consequences on muscle size and protein synthesis [43,83,84]. In particular, by using skeletal muscle-specific, inducible Raptor knockout mice, it was recently demonstrated that mTORC1 can either attenuate or augment denervation-induced atrophy, depending on the fiber type [81], while mTORC2 inhibition does not affect neurogenic muscle atrophy [80]. Thus, depending on the net effect between protein synthesis and degradation and the fiber-type context, mTORC1 activation could result in spared or exacerbated muscle atrophy.

## 2. Functional Foods to Ameliorate Neurogenic Muscle Atrophy

Functional foods are considered as either fresh or processed foods that naturally promote health and prevent or ameliorate diseases beyond their basic nutritional functions [85]. Functional foods or ingredients with protective, therapeutic outcomes against neurogenic muscle atrophy have recently received increasing consideration, being suitable for long-term treatment and potentially free from side effects. Here we provide an overview of the foods known to be of benefit for counteracting neurogenic muscle atrophy. Whenever possible, the findings are discussed within a comparison with age-related sarcopenia, considering results from animal models and human studies.

### 2.1. Protein or Amino Acids

High daily protein intake (1.0–1.2 g/kg/day) correlates with the maintenance of muscle mass with aging [86]. Such assumption persists even though contrasting data were reported from randomized clinical trials employing protein or amino acid supplements [87,88,89,90,91], probably due to the heterogeneity of the initial populations and to the difficulty in measuring the effective protein intake throughout the studies. Similarly, a positive correlation has been found between the length of survival of ALS patients and the amount of protein consumed, especially at the early stage of the disease [92]. However, dietary supplementation with creatine or amino acids has not been proven to be effective in improving the prognosis of ALS patients [93]. 

When the studies are stratified by the specific amino acids used, the most interesting results are the following (Table 1):Branched-chain amino acids (BCAAs), i.e., leucine, isoleucine, and valine, are three of the nine essential amino acids and are found in protein-rich foods such as eggs, meat, and dairy products. A BCAA-enriched diet has been shown to be protective against sarcopenia in both slow and fast muscles of old mice [94] and in elderly subjects [89,95,96], by promoting mitochondrial genesis, thus improving muscle endurance, and decreasing oxidative stress. In contrast, treatment of ALS patients with BCAAs or L-threonine for six months failed to show beneficial effects on disease progression [97].Leucine is an essential BCAA present in all protein-rich foods, but is more abundant in those of animal origin [98]. A leucine-enriched exclusive diet prevented neurogenic muscle atrophy in denervated rat soleus (slow) muscle, by increasing the AKT/mTOR anabolic pathway and decreasing the AMPK-catabolic one [99]. Leucine supplementation is potentially useful to increase protein synthesis for counteracting muscle sarcopenia in elderly subjects [100,101].

As for the mechanism underlying their beneficial effects, the dietary intake of essential amino acids increases the expression of several miRNAs in the skeletal muscle, including miR-1, miR-23a, miR-208b, miR-499, and miR-27a, regulating the expression of muscle-specific growth-related genes and thereby promoting muscle mass [102,103]. 

Beta-Hydroxy-Beta-Methyl Butyrate (HMB) is a natural metabolite of the amino acid leucine, and is found in small quantities in grapefruit, alfalfa, and catfish. In humans, 2–10% of dietary L-leucine is converted to HMB, corresponding to about 0.3 g/day; however, when taken as a supplement the doses are 10–20-fold higher [104]. As a derivate of a BCAA, HMB is thought to act as pro-anabolic and anti-catabolic compound for skeletal muscle, and early studies showed a positive impact of HMB supplementation in counteracting the age-related losses of skeletal muscle mass in elderly subjects [105,106,107]. However, two recent articles published opposite results: according to a review, the current evidence is inconclusive with regard to any positive effects of HMB supplementation on functional outcome measurements or muscle mass in elderly human subjects and in hospitalized patients [108], while a meta-analysis concluded that HMB supplementation helps increase muscle strength in elderly people [109]. To the best of our knowledge, no studies on HMB supplementation in animal models of neurogenic muscle atrophy are currently available. Studies are needed to clarify the pathways hit by HMB in denervated or aged skeletal muscles, prior to proposing the use of HMB for counteracting neurogenic muscle atrophy or sarcopenia.Creatine is a compound derived from glycine and arginine which is mostly present in skeletal muscle, where it is used as an energy store, and can be found in red meat and seafood. Abundant evidence indicates that creatine supplementation increases skeletal muscle mass and strength if associated with resistance training [110], due to its beneficial effects on decreasing muscle protein breakdown, inflammation, and oxidative stress; interestingly, this also holds true with aging [111,112]. Despite the positive effects on skeletal muscle homeostasis, current literature suggests that exogenous creatine supplementation appears to be poorly effective in treating ALS [113,114].Carnitine is a quaternary ammonium compound required for the transport of long-chain fatty acids into mitochondria for energy production. Carnitine can be mainly found in animal products such as meat, fish, poultry, and milk, and is involved in skeletal muscle protein homeostasis by regulating both protein synthesis and breakdown, being an antioxidant and anti-inflammatory compound [115]. Interestingly, denervation and aging decrease carnitine levels in both slow and fast rat skeletal muscles [116], suggesting a causative role for carnitine in the atrophic program. Carnitine supplementation, alone or in combination with physical exercise, counteracts the age-dependent decline of mitochondrial function in the soleus rat muscle, improving muscle energy production and body protein mass [117,118,119]. The positive effects of carnitine supplementation were also reported in a murine model of ALS [120], as well as in a phase II clinical trial [121], proving to be effective in slowing down the progression of muscle weakness and prolonging mouse and patient survival.Carnosine is a dipeptide composed of the beta-alanine and histidine amino acids and mainly present in meats. Carnosine exerts numerous positive actions in skeletal muscle, including antioxidant and antiglycation activity, enhanced calcium sensitivity, and H^+^ buffering [122] that may affect muscle performance and maintenance in aging and neuromuscular diseases [123,124,125]. In elderly subjects, an increase in carnosine intake counteracted cognitive decline and improved physical capacity, probably due to its anti-inflammatory action [126,127]. Further studies are needed to clarify the effects of carnosine supplementation on skeletal muscle proteolysis and synthesis in a denervation-dependent condition such as in aging or ALS.

**Table 1 metabolites-12-01149-t001:** Amino acids or peptides able to counteract neurogenic muscle atrophy.

Functional Nutrient	Disease or Condition	Experimental Model	Mechanism of Action	References
BCAAs	Sarcopenia	Aged mice	- Promote mitochondrial formation and function - Decrease oxidative stress	[94]
		Elderly human subjects		[95,96,128]
Leucine	Nerve rescission	Rat	- Increases the AKT/mTOR - Decreases the AMPK catabolic pathways	[99]
	Sarcopenia	Elderly human subjects	- Improves lean muscle-mass content	[100,101]
Creatine	Sarcopenia	Adults human subjects	- Decreases muscle protein breakdown, inflammation, and oxidative stress	[110,111,112]
Carnitine	Sarcopenia	Rat	- Increases mitochondrial function and muscle mass	[117,118,119]
	ALS	Mice and human patients	- Delays ALS onset and progression, prolongs survival	[120,121]
Carnosine	Aging	Elderly human subjects	- Reduces OS, inflammation, inhibits protein glycation and aggregation	[126,127]

### 2.2. Lipids

Unsaturated fatty acids have been reported to protect against muscle wasting in response to various pathological conditions. In particular, the replacement of saturated fatty acids (SFAs) by either mono- (MUFAs) or poly-unsaturated (PUFAs) has been associated with a lower sarcopenia risk score in a regression analysis that involved almost a thousand participants [129]. This study confirmed what has already been reported by others, which is the ability of unsaturated fatty acids to improve muscle mass and function in elderly subjects [86,130]. Interestingly, not all unsaturated fatty acids are the same, at least for ALS disease. Indeed, a higher intake of omega-3 PUFAs, but not omega-6, inversely correlates with ALS risk [131], while vitamin E consumption, in addition to omega-3, reinforces this inverse correlation [132].

Several possible signaling pathways activated by PUFAs may underpin the beneficial effects on counteracting neurogenic muscle atrophy. Indeed, PUFAs exert a well-recognized anti-inflammatory action, increase nerve conduction, improve mitochondrial function, and stimulate muscle protein synthesis [133,134] while modulating autophagy and countering protein degradation [86,135].

We report the effects for the most promising lipids to treat muscle atrophy in Table 2. They are discussed in detail below:Linoleic acid (LA) is an omega-6 PUFA that is found in vegetable oils, nuts, seeds, meats, and eggs [136]. LA was already known for its beneficial effects on skeletal muscle cells, demonstrated in vitro [137] and in vivo against muscular dystrophy [138]. Very recently, it has also been shown that LA treatment counteracts neurogenic muscle atrophy in mice by preventing the denervation-induced increase of oxidative stress and UPS [139].Fish oil is available from many types of fish and shellfish. It is rich in two important omega-3 fatty acids, i.e., eicosapentaenoic acid (EPA) and docosahexaenoic acid (DHA). A fish oil-enriched diet has been shown to be protective against neurogenic muscle atrophy in mice by suppressing the TNF-α-dependent UPS catabolic pathway [140]. Interestingly, a therapy based on fish oil-derived n3 PUFA has been proposed as a natural approach in a clinical trial to counteract sarcopenia, being partially effective in preventing the decline of muscle volume and strength in elderly subjects [134]. Jeromson et al. [141] stress that n-3 supplementation in humans increases sensitivity to anabolic stimuli and shows additional, anticatabolic effects. While EPA has been shown to antagonize the actions of TNF-α on C2C12 myotube formation, it also diminishes the activation of the NF-κB pathway, thereby reducing MuRF1 signaling in these cells. EPA, in addition, favorably affects cell metabolism in the face of alterations in the availability of the different energy substrates, promoting plasticity, and improves glucose uptake in cultured myotubes, as reported in [141].

Another aspect to which omega-3-fatty-acids may contribute to mitigation of atrophy is by beneficially modulating neuromuscular function, playing a suggested role on both the nerve and on the muscle fiber, altering membrane fluidity and sensitivity to acetylcholine (reviewed by [142]).

Medium-chain triglycerides (MCT) are six-twelve carbon fatty acid esters of glycerol that can be found in coconut oil and products, palm oil, and dairy products. In mice, an MCT-enriched diet triggers glucose and lipid metabolism pathways and mitochondrial biogenesis in the skeletal muscle, thus improving muscle function under high-temperature conditions [143]. MCTs alone, or in combination with leucine and vitamin D, increase muscle strength and function in elderly adults [144,145]. However, MCT administration is detrimental for the heart of healthy or dystrophic mice [146,147]. Thus, careful examination of the systemic effects of MCT administration is encouraged in clinical studies.Alkylresorcinols (ARs), differently from PUFA, are amphiphilic phenolic lipids present in many kinds of cereals. Interestingly, AR dietary supplementation prevented neurogenic muscle atrophy in mice by affecting the autophagic pathway and thereby the lipid metabolism, but not the UPS activation [148]. This finding is intriguing, confirming that challenging metabolism may be an efficient therapeutic approach for maintaining muscle homeostasis, as suggested in ALS [149] and already described in other muscle pathologies [35,150,151].

**Table 2 metabolites-12-01149-t002:** Lipids effective in counteracting neurogenic muscle atrophy.

Functional Nutrient	Disease or Condition	Experimental Model	Mechanism of Action	References
Omega-3 fatty acids	Sarcopenia	Elderly human subjects	- Stimulates muscle protein synthesis via mTOR	[133]
Fish oil-derived omega-3 fatty acids	Sarcopenia	Elderly human subjects	- Increases muscle mass and function	[134]
Omega-3 fatty acids	ALS	ALS patients	- Prevents or delays ALS onset	[131]
Linoleic acid	Nerve Rescission	Mice	- Counteracts the increase of oxidative stress and UPS activation	[139]
Fish oil	Nerve Rescission	Mice	- Suppresses the UPS activation	[140]
MCTs		Elderly human subjects	- Increases muscle strength and function	[144]
Alkylresorcinols	Nerve Rescission	Mice	- Modulates autophagy	[148]

### 2.3. Vitamins

Vitamins are essential micronutrients that cannot be synthesized by the organism, therefore, they must be obtained via the diet. Vitamins prevent skeletal muscle atrophy mainly for their anti-inflammatory properties and by buffering oxidative stress, thus affecting both protein synthesis and catabolism [152]. As for neurogenic muscle atrophy, however, uncertain findings have been reported on the effectiveness of vitamin supplementation (Table 3).

β-carotene is a red-orange pigment found in fruits and vegetables. The human body converts β-carotene into vitamin A. β-carotene attenuates ROS-dependent muscle atrophy in C2C12 myotubes by repressing the activation of atrogin-1 and MuRF1 [153]. In the elderly, high plasma concentrations of β-carotene and other similar antioxidants correlate with preserved muscle function [154], which would support their clinical use. However, weak data were reported about the effectiveness against neurogenic muscle atrophy in vivo, as β-carotene supplementation is only effective at the early stage of soleus muscle atrophy upon denervation [155].As for vitamin C, both dietary and circulating levels positively correlate with skeletal muscle mass measurements in middle- and older-aged subjects, suggesting that dietary vitamin C intake may be protective against sarcopenia [156]. Coherently, mice with defective vitamin C biosynthesis (SMP30-knockout mice) develop muscle atrophy in both fast and slow muscles, with high expression of muscle-specific E3-ubiquitin ligases, atrogin-1 and MuRF1, and high levels of ROS. Vitamin C supplementation was able to recover the SMP30-knockout muscle atrophy phenotype [157], highlighting its direct involvement in the maintenance of skeletal muscle homeostasis.Vitamin D certainly plays an important role in skeletal muscle physiology. Vitamin D deficiency in humans leads to muscle weakness and myalgia that can be reverted by vitamin D replenishment [158]. Vitamin D supplementation is associated with a significant increase in muscle mass and function in older adults with sarcopenia, especially for those with a significant baseline vitamin D deficiency [159,160,161,162]. At the molecular levels, skeletal muscle seems to express the vitamin D receptor, which mediates vitamin D-dependent signaling affecting calcium handling [163]. In addition to beneficial effects on age-related sarcopenia (revised in [164]), vitamin D helps muscle recovery following strenuous muscular activity. However, when vitamin D was supplemented in ALS, negative results prevailed over beneficial findings [165,166,167,168].Trolox, the cell-permeable derivative of vitamin E, was not able to exert any protective effects on neurogenic muscle atrophy in mice [62], despite evidence that the combined supplementation of whey protein, vitamin D and E can significantly improve muscle mass, strength, and markers of protein anabolism in sarcopenic subjects [169]. Two studies acknowledged the importance of oxidative stress in neurogenic muscle atrophy: the divergent conclusions come from different treatments (single or combined) and different models (mice vs. humans).

In conclusion, vitamins in general, mainly vitamin B12, E, and C, are the most significant protectors inhibiting ALS development [170], mainly due to their neuroprotective effects rather than their effects on muscle mass loss modulation. More preclinical studies, confirmed by human trials, are required to better define the molecular pathways and the effectiveness of vitamin supplementation to counteract skeletal muscle atrophy following denervation. 

**Table 3 metabolites-12-01149-t003:** Vitamins affecting neurogenic muscle atrophy.

Functional Nutrient	Disease or Condition	Experimental Model	Mechanism of Action	References
Beta-carotene	Nerve rescission	Mice	- Represses the UPS activation	[155]
Vitamin C	Sarcopenia	SMP30-KO mice	- Hampers the UPS activation	[157]
Vitamin D	Sarcopenia	Elderly human subjects	- Improves muscle mass and strength	[159,160,161,162]
Whey protein + vitamin D + vitamin E	Sarcopenia	Elderly human subjects	- Increase muscle mass, muscle strength, and anabolic markers	[169]

### 2.4. Plant-Derived Ingredients

Several plant-derived ingredients prevent or reduce muscle atrophy, especially by inhibiting muscle protein degradation while enhancing muscle synthesis, and/or exerting anti-inflammatory and anti-oxidative functions [171,172]. We describe below and summarize in Table 4 the most effective ones.

Geranylgeraniol (GGOH), is a plant-derived isoprenoid with some beneficial effects for skeletal muscle mass. GGOH administration reduced the loss of myofiber size upon denervation in rat gastrocnemius muscle by affecting the expression of atrogin-1 [173], without enhancing muscle growth, even though it had previously shown a positive effect on C2C12 myoblast differentiation in vitro [174]. In addition, GGOH may interfere with the NF-κB and/or testosterone signaling in skeletal muscle, as demonstrated in other cell types [174,175], thus contributing to the protection against neurogenic muscle atrophy.Capsaicin is a chili pepper-derived extract with analgesic properties. Its protective action against neurogenic muscle atrophy has been described in a study [176]. Capsaicin administration, used as an agonist of the transient receptor potential cation channel, subfamily V, member 1 (TRPV1), induced muscle hypertrophy and alleviated denervation-induced atrophy in both fast and slow murine muscles. Activated TRPV1 increased intracellular Ca^2+^ concentration leading to mTOR activation and muscle biosynthesis [176].Polyphenols are a wide group of plant-derived organic compounds found in fruits, vegetables, coffee, tea, and whole grains. They are believed to be potential therapeutic agents for inhibiting muscle atrophy and improving muscle mass and strength [172]. Polyphenols act primarily as antioxidants and anti-inflammatory agents, thus inhibiting muscle atrophy-related genes and promoting the activation of the IGF-1 signaling pathway [172,177]. In addition, in vitro and in vivo observations proved the neuroprotective effects of these bioactive compounds by improving mitochondrial biogenesis and function, reducing toxic protein aggregates and microglia and astrocytes inflammation, and overall favoring motor neuron survival [178]. Some examples are reported below.

Isoflavones are a type of polyphenol mainly found in legumes, such as soybeans and other fruits and nuts. The administration of isoflavones has been shown to counteract TNF-α-induced C2C12 myotube atrophy by reducing MuRF1 promoter activity, and to prevent neurogenic muscle atrophy in vivo, when mice were pre-treated with isoflavones before denervation, mainly by interfering with apoptosis-dependent signaling [179,180].

Curcumin is extracted from the rhizome of turmeric, from the ginger family of plants, and displays anti-aging properties [181], by increasing the activity of SOD enzyme and increasing lifespan [182]. Accordingly, curcumin counteracts sarcopenia in mice [183], and clinical trials with healthy elderly subjects are ongoing [184]. Since curcumin is an antioxidant, as also reported above [185], it would be an ideal candidate as a nutraceutical for ALS and other neurodegenerative diseases. Indeed, a curcumin derivative, GT863, has been proven to slower motor dysfunction in a murine model of ALS [186]. In a different study exploiting the same SOD1 model, curcumin reduced the cytotoxicity of the amylogenic pathway [187]. Interestingly, curcumin protects the peripheral nervous system after injury [188] and promotes nerve regeneration [189]. The clinical use of curcumin in neurological disorders has been reviewed recently [190].

Resveratrol is a polyphenol found in grapes, red wine, and berries. It has been shown to inhibit protein degradation and protect skeletal muscle against atrophy in different in vivo models, including cancer, diabetes, chronic kidney disease, and disuse [191,192,193,194]. Dietary resveratrol supplementation prevented neurogenic muscle atrophy in mice, likely by interfering with the activation of UPS and autophagic pathways [195], in addition to its neuroprotective functions [196]. Moreover, resveratrol affects muscle mass by promoting muscle cell differentiation and PGC-1α activation through the concomitant up-regulation of miR-21 and miR27-b and downregulation of miR-133b, miR-30b, and miR-149 [197]. Convincing data on the effects of resveratrol on muscle mass in experimental models of neurodegenerative diseases are still missing.

Avenanthramides (AVNs), also known as N-cinnamoylanthranilate alkaloids or anthranilic acid amides, are a group of low molecular weight phenolic amides found mainly in the whole grain oat [198]. AVNs are effective in inhibiting NF-kB activation, ROS production, and proinflammatory cytokine expression in several cell types, including muscle cells [199,200,201]. In particular, AVNs bind and inhibit IKKβ activity, thereby inhibiting the NFκB-mediated inflammatory response in muscle cells [202] and TNF-α-induced myotube atrophy [199].

Among polyphenolic compounds, flavonoids are a subgroup generally having a 15-carbon skeleton. Both polyphenols and flavonoids exert anti-inflammatory and antioxidant actions and some flavonoids have been reported to possess protective functions, specifically with regard to neurogenic muscle atrophy.

Quercetin is a flavonoid found in many fruits and vegetables, such as in red wine, onions, green tea, apples, and berries. Quercetin has been shown to play protective functions against the muscle wasting that accompanies a variety of conditions. Regarding neurogenic muscle atrophy, quercetin suppresses TNF-α-induced C2C12 myotube atrophy by interfering with the activation of NF-κB and the consequent activation of the ubiquitin ligases atrogin-1 and MuRF-1, and by activating the Heme Oxygenase 1 (HO-1) [203]. Moreover, the administration of quercetin before denervating mice prevents neurogenic muscle atrophy by increasing mitochondriogenesis and function via PGC-1α [204].

Epicatechin is one of the most abundant flavonoids present in different fruits and green tea. Dietary supplementation of epicatechin prevents muscle loss in sarcopenic mice by increasing the expression of the myogenic marker MyoD and the antioxidant stress-related enzymes SOD and catalase, while reducing the expression of the catabolic marker genes, such as FoxO3a, myostatin, and MuRF1 [205]. Similarly, dietary supplementation with another green tea polyphenol, i.e., epigallocatechin-3-gallate, increases skeletal muscle mass and size in aged rats by downregulating the expression of the E3-ubiquitin ligases, MuRF1 and atrogin-1, and myostatin, and by increasing IGF-1 [206]. 

Apigenin is a flavonoid found in plants such as parsley, celery, and grapefruit. An apigenin-supplemented diet was able to prevent neurogenic muscle atrophy in both fast and slow murine muscles by inhibiting denervation-induced MuRF1 and IL-6 expression, thus affecting the activation of UPS and inflammatory processes within the muscle [207].

Genistein is a flavonoid derived from legumes. Dietary genistein supplementation attenuated the denervation-induced muscle atrophy in slow muscles in mice by interfering with the estrogen receptor-mediated activation of MuRF1 and atrogin1 expression [208].

8-prenylnaringenin is a prenylated flavonoid found in hops. Dietary ingestion of 8-prenylnaringenin prevents neurogenic muscle atrophy in mouse gastrocnemius muscles by stimulating the activation of Akt phosphorylation and preventing the induction of the key ubiquitin ligase involved in muscle atrophy atrogin-1 [209]. 

Tomatidine is the metabolite obtained from α-tomatine, a glycoalkaloid abundantly present in tomato plants. Strikingly, the mRNA expression signature of tomatidine negatively correlates to that one of skeletal muscle atrophy upon fasting and spinal cord injury [210], suggesting that tomatidine might exert an anti-atrophic effect on skeletal muscle. Indeed, tomatine administration induced functional muscle hypertrophy, both in vitro and in vivo, accompanied by reduced adiposity, by activating mTORC1 signaling [210]. By enhancing protein synthesis and mitochondriogenesis, tomatidine also prevented muscle atrophy induced by fasting or immobilization in mice [210]. In *C. elegans*, tomatidine improves muscle function during aging by activating mitophagy and antioxidant cellular defenses [211]. Based on these promising findings, the use of tomatidine to counteract neurogenic muscle atrophy should be further investigated, along with the delineation of the molecular mechanisms underpinning the atrophy rescue.Tinospora cordifolia is a plant found in tropical and sub-tropical parts of Asia, Africa, and Australia, whose extract (TCE) has been widely used in ancient Ayurvedic and Tibetan medicine due to its high antioxidant activity [212]. TCE supplementation prevented neurogenic muscle atrophy by enhancing protein synthesis by antagonizing the proteolytic pathways (calpain and UPS), and by enhancing the oxidative stress response in both slow and fast mouse muscles [213].Salidroside is a glucoside of tyrosol extracted from the plant Rhodiola rosea with anti-inflammatory, anti-oxidative, and anti-apoptotic properties [214,215]. Salidroside protects skeletal muscle from neurogenic muscle atrophy due to its anti-inflammatory properties [216].

While significant evidence was collected regarding the protective effects of plant-derived ingredients, the molecular mechanisms of most of them have been examined only in preclinical studies, either in vitro or in vivo. The clinical data are still missing and are urgently needed. In addition, most of the studies are correlative, presenting associations between the use of dietary supplementation and the muscle phenotype. Little evidence supports the structure-function relationship of phytochemicals in skeletal muscle atrophy, which needs to be investigated further.

**Table 4 metabolites-12-01149-t004:** Plant-derived ingredients effective on neurogenic muscle atrophy.

Functional Nutrient	Disease or Condition	Experimental Model	Mechanism of Action	References
Geranylgeraniol	Nerve rescission	Rats	- Interferes with the UPS activation	[173]
Capsaicin	Nerve rescission	Mice	- Increases [Ca^2+^]_i_ leading to mTOR activation and muscle biosynthesis	[176]
Isoflavones	TNF-α-induced muscle atrophy Nerve rescission	C2C12 myotubes Mice	- Interferes with the activation of MuRF1 - Interferes with the apoptosis-dependent signaling	[179,180]
Curcumin	ALS	Mice ALS patients	- Decreases amyloid formation - Diminishes oxidative stress	[187,217]
Resveratrol	Nerve rescission	Mice	- Blunts the UPS and autophagy activation	[195]
	Sciatic nerve crush injury	Rats	- Neuroprotective functions	[196]
AVNs	TNF-α-induced muscle atrophy	C2C12 cells	- Inhibits NF-kB activation, ROS production, and proinflammatory cytokine expression	[199,202]
Quercetin	TNF-α-induced muscle atrophy Nerve rescission	C2C12 myotubes Mice	- Inhibits NF-kB activation - Activates HO-1 - Increases mitochondriogenesis and function	[203,204]
Epicatechin	Sarcopenia	Aged mice	- Increases protein synthesis - Improves the oxidative stress response - Prevents UPS activation	[205]
Epigallocate-chin-3-gallate		Aged rats	- Prevents UPS activation - Reduces myostatin expression - Increases the anabolic pathway	[206]
Apigenin	Nerve rescission	Mice	- Inhibits UPS activation - Reduces inflammation	[207]
Genistein	Nerve rescission	Mice	- Prevents UPS activation	[208]
8-prenylnaringenin	Nerve rescission	Mice	- Activates the AKT anabolic pathway - Interferes with UPS activation	[209]
Tomatidine	Sarcopenia	C. elegans	- Activates mitophagy - Reduces oxidative stress	[211]
Tinospora cordifolia	Nerve rescission	Mice	- Enhances protein synthesis - Antagonizes the proteolytic pathways - Increases the oxidative stress response	[213]
Salidroside	Nerve rescission	Rat	- Anti-inflammatory properties	[216]

### 2.5. Prebiotics, Probiotics and Dietary Fibers

Skeletal muscle wasting conditions, in addition to aging, deeply affect the intestinal mucosa [218] and are associated to changes in the gut biota [219,220]. Vice-versa, the gut microbiota can potentially affect skeletal muscle mass by modulating systemic inflammation and immunity, energy metabolism, and insulin sensitivity [221]. The importance of gut microbiota for the maintenance of skeletal muscle physiology and homeostasis has been recently demonstrated by comparing germ-free and pathogen-free mice [222]. The former, lacking a gut microbiota, showed skeletal muscle atrophy with clear molecular signs of muscle denervation, a phenotype that was reversed by transplanting the gut microbiota of the pathogen-free mice. Transplantation of the gut microbiota from pathogen-free mice into germ-free mice affects both slow and fast muscles, reducing the skeletal muscle atrophy markers atrogin-1 and MuRF1, and increasing mitochondriogenesis and oxidative metabolism, thus restoring skeletal muscle mass [222]. Additional studies have directly or indirectly proved a certain relationship between muscle mass and gut microbiota [223,224,225]. Although the pre and probiotics’ mechanisms for rescuing muscle mass are not yet well defined, gut microbiota may potentially influence muscles via endocrine and insulin sensitivity, energy metabolism, immunity, and inflammation [226]. Microbiota composition can be modified with the diet by increasing the consumption of prebiotics, i.e., non-digestible carbohydrates fermented in the lower part of the gut that stimulates the growth and/or activity(ies) of bacteria, or probiotics, i.e., live microorganisms, thus conferring a health benefit on the host. Often prebiotics have pleiotropic effects, regulating metabolism while acting on fat tissue and lean mass [227]. 

Since a dietary intervention alters the gut microbiota, it may be used as a potent tool to heal elderly subjects [228]. Indeed, dietary fiber administration is protective against age-related muscle loss by improving glucose metabolism, muscle function and lean body mass in adult subjects [229,230]. Such positive effects on skeletal muscle maintenance are attributed to dietary fiber’s ability to rapidly and reproducibly change the composition of microbiota, i.e., living members forming the microbiome, and microbiome, i.e., all of the genetic material within a microbiota [231,232,233]. For example, high dietary fiber intake increases the gut microbiota production of short-chain fatty acids [231], which are important regulators of skeletal muscle mass, metabolism, and function [234]. Beneficial effects of fiber supplementation have also been reported in metabolic disease conditions due to the decrease of insulin resistance [235,236] and pro-inflammatory cytokine concentration [237,238], which are two mechanisms directly involved in sarcopenia. In keeping with the approach of affecting microbiota, the administration of a prebiotic composed of a mixture of inulin and fructooligosaccharides for 13 weeks increased some skeletal muscle functional parameters in a randomized controlled double-blind study in elderly people [239].

Similarly to aging, the gut microbiota varies between ALS patients and healthy subjects; moreover, it changes further during the progression of the disease, with a decrease of potentially protective microbial groups, such as Bacteroidetes, and an increase of those groups with potential neurotoxic or pro-inflammatory activity, such as Cyanobacteria [240]. Supplementation with a probiotic formulation, consisting of a mixture of five lactic acid bacteria, for 6 months, despite regulating the intestinal microbiota of patients, did not influence the progression of the disease, as evaluated by ALS Functional Rating Scale–Revised (ALSFRS-R) score [240]. Still, we find these findings as potentially false-negative results, considering that out of 50 patients enrolled in the study, 20 discontinued the study before the 6-month follow-up period, and one patient died. The limited sample size is one of the main limits to be overcome for the studies in the field of the importance of the microbiota in ALS, in addition to the reluctance of neurologists to study ALS as a multiorgan syndrome other than limited to the nervous system. 

## 3. Conclusions

A specific and personalized dietary intervention might be routinely administered as a therapeutic approach, in combination with a pharmaceutical intervention and exercise, to counteract the burden of neurogenic muscle atrophy. Such a safe approach should be also considered to slow down the age-dependent muscle wasting in sarcopenia. Preclinical studies, especially those involving animal models of muscle denervation, are effective in better elucidating the molecular mechanisms underlying the functional food-mediated rescue of neurogenic muscle atrophy (Figure 2); in this regard, the potential role of the miRNAs as novel candidates for mediating the food effects on muscle mass [241,242] pave the way to a new avenue for investigation. Further research on functional nutrients is needed to support the development and design of precision medicine strategies.

## Figures and Tables

**Figure 1 metabolites-12-01149-f001:**
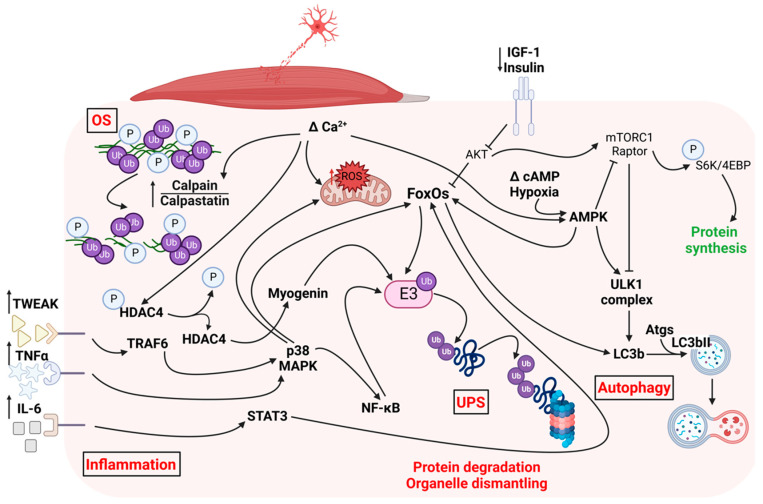
Pathways triggered by denervation in the skeletal muscle. In red are mentioned that one promoting muscle atrophy, while in green the one preserving muscle mass. OS, oxidative stress; UPS, ubiquitin-proteasome system; P, phosphate group, E3, E3-ubiquitin ligase; Ub, ubiquitin; Δ, changes.

**Figure 2 metabolites-12-01149-f002:**
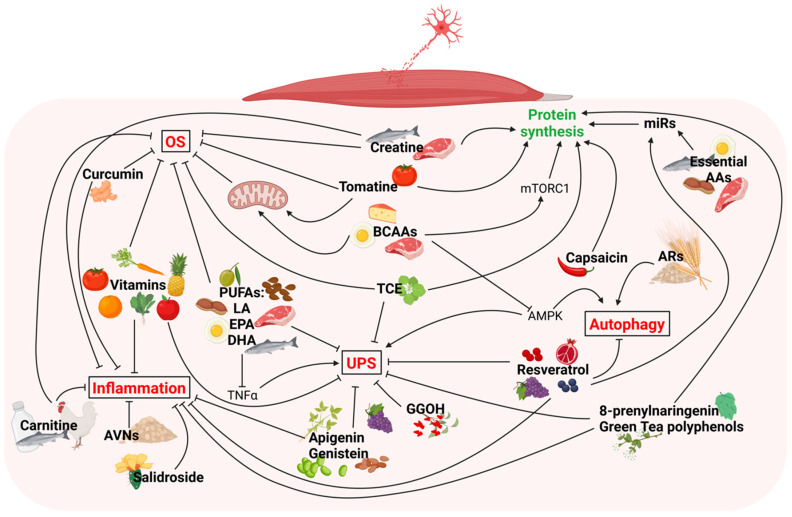
Functional nutrients able to prevent denervation-induced skeletal muscle atrophy by decreasing muscle catabolism or increasing muscle anabolism. In red are mentioned the pathways promoting muscle atrophy, while in green the one preserving muscle mass.

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
