# Peer review of "Functional Nutrients to Ameliorate Neurogenic Muscle Atrophy"

_metabolites, 2022, doi:10.3390/metabo12111149_

Round 1
Reviewer 1 Report
This manuscript entitled “Functional nutrients to ameliorate neurogenic muscle atrophy” focused on the functional support to ameliorated neurogenic muscle atrophy. And the functions of food ingredients on regulating activities of the ubiquitin-proteasome pathways, the autophagic/lysosomal pathway, and calpains were described. The potential mechanisms of food/plant-derived peptides, lipids, vitamins, and phytochemicals on attenuating neurogenic muscle atrophy. The manuscript is interesting; however, there are still some major concerns.
1. The relationships of food nutrients such as protein/peptides, lipids, and vitamins and neurogenic muscle atrophy have been revealed. The functions of some other potential nutrients like dietary fiber on attenuating muscle atrophy should be included. Although the author had described the protective ability of pre- and probiotics on muscle healthy maintaining, the correlation of dietary fiber on regulating gut microbiota and muscle mass.
2. The novel phytochemicals should be included in this review. Avenanthramides had been reported to maintain muscle mass, the articles such as PMID: 30997266, PMID: 34994561 and potential function of avenanthramides on regulating muscle atrophy should be added in the section of “Plant-derived ingredients”.
It is strongly encouraged to add some potential new mechanisms of food ingredients on regulating muscle atrophy. At least, the potential new signal pathway should be mentioned for the prospective direction of future research. And the miRNAs have been considered to be a novel candidate for food ingredients regulating muscle mass. The latest publications such as “MicroRNA: the novel mediators for nutrient-mediating biological function” and “Sarcopenia: Etiology, Nutritional Approaches, and miRNAs.” should be referred.
Reviewer 2 Report
The paper titled 'Functional nutrients to ameliorate neurogenic muscle atrophy' provides a comprehensive overview of the role of nutrients in an often-neglected area of research; neurogenic muscle atrophy. The authors provide a summary of conditions and pathways triggered by skeletal muscle before discussing nutrients that may be effective in counteracting neurogenic muscle atrophy. The topic is interesting and understudied, and may provide support to current therapeutic approaches. The authors supported their review of evidence with animal experimental models and, also, human studies, to allow the reader to easily comprehend the extent of the effect of nutrients, if any. The paper is well-written and easy to read, and I could see it as a reference to postgraduate students interested in this area of research. In their display of evidence, the authors highlight the areas of research with contradictory conclusions that need further investigation, which can help guide future researchers. In the conclusion, the authors highlight the main findings within the context of their review and carefully note the role of dietary intervention as a possible therapeutical approach without editorializing.Author Response
Please see the attachment.

Reviewer 3 Report
The authors describe the various pathways activated by skeletal muscle denervation leading to muscle atrophy. This area of research is critical due to its physiological similarities with age-related skeletal muscle loss, sarcopenia, muscle wasting in cancer cachexia, and other diseases like ALS and MS. The review of the signaling pathways activated by skeletal muscle denervation gives an overview of the interactions and synergies leading to muscle atrophy and dysfunction. The authors listed several functional food ingredients and natural bioactives that can counter skeletal muscle atrophy by either enhancing anabolic pathways (AKT/mTOR) or inhibiting catabolic pathways (UPS, Autophagy, protein degradation). I would recommend the authors the suggestions below:
1- In Figure show using arrow intracellular messenger and pathways activated by muscle denervation using arrows, for example. The pathological increase in cytosolic calcium concentrations increased calpain activity after muscle denervation.
2- The review overlooked the role of Calpastatin, a key negative allosteric regulator of calpain activity. The authors should explore the role of Calpastatin downregulation and calpain upregulation after muscle denervation or aging, thus contributing to skeletal muscle atrophy and catabolism.
3- IRS-1/PI3K/AKT/mTOR signaling axis that promotes protein synthesis and muscle (anabolic pathways) should be put in clear contradiction to the other catabolic pathways in Figure 1)
4- The authors disregarded the contribution of systemic chronic inflammation (TNF-a, NFkB pathways) and idiopathic inflammatory myopathy in the development of muscle atrophy and muscle wasting. It seems to be more relevant than oxidative stress (OS), which is stressed throughout the manuscript. Several of the bioactives in functional food (prebiotic/probiotic, polyphenols, etc.) have known anti-inflammatory properties.
5- The paragraph about polyphenols should be more structured. It should be evident to the reader that the list of polyphenols and flavonoids listed underneath are still within the same paragraph
6- Edit reference citation put in lines 311-316 as appropriate
7- Reference is missing for the cited review in Line 320
8- Please correct “(such as fructans)” in line 500. Fructans are fermentable prebiotic fibers described in line 499
Round 2
Reviewer 1 Report
All the comments had been well addressed.
Author Response
We have addressed all reviewer concerns
Reviewer 3 Report
The authors have made significant improvements to the manuscript and have addressed the major comments.
Author Response
We have addressed all reviewer concerns